# Mind the gap: What explains the poor-non-poor inequalities in severe wasting among under-five children in low- and middle-income countries? Compositional and structural characteristics

**Adeniyi Francis Fagbamigbe**[1,2¤]*, **Ngianga-Bakwin Kandala**[3], **Olalekan A. Uthman**[2]

**1** Department of Epidemiology and Medical Statistics, College of Medicine, University of Ibadan, Ibadan, Nigeria, **2** Division of Health Sciences, Populations, Evidence and Technologies Group, University of Warwick, Coventry, United Kingdom, **3** Department of Mathematics, Physics & Electrical Engineering (MPEE), Northumbria University, Newcastle, United Kingdom

¤ Current address: Department of Epidemiology and Medical Statistics, College of Medicine, University of Ibadan, Nigeria
* franstel74@yahoo.com

**Data Availability Statement:** All data is freely available at http://dhsprogram.com. The authors

## Abstract

A good understanding of the poor-non-poor gap in childhood development of severe wasting (SW) is a must in tackling the age-long critical challenge to health outcomes of vulnerable children in low- and middle-income countries (LMICs). There is a dearth of information about the factors explaining differentials in wealth inequalities in the distribution of SW in LMICs. This study is aimed at quantifying the contributions of demographic, contextual and proximate factors in explaining the poor-non-poor gap in SW in LMICs. We pooled successive secondary data from the Demographic and Health Survey conducted between 2010 and 2018 in LMICs. The final data consist of 532,680 under-five children nested within 55,823 neighbourhoods from 51 LMICs. Our outcome variable is having SW or not among under-five children. Oaxaca-Blinder decomposition was used to decipher poor-non-poor gap in the determinants of SW. Children from poor households ranged from 37.5% in Egypt to 52.1% in Myanmar. The overall prevalence of SW among children from poor households was 5.3% compared with 4.2% among those from non-poor households. Twenty-one countries had statistically significant pro-poor inequality (i.e. SW concentrated among children from poor households) while only three countries showed statistically significant pro-non-poor inequality. There were variations in the important factors responsible for the wealth inequalities across the countries. The major contributors to wealth inequalities in SW include neighbourhood socioeconomic status, media access, as well as maternal age and education. Socio-economic factors created the widest gaps in the inequalities between the children from poor and non-poor households in developing SW. A potential strategy to alleviate the burden of SW is to reduce wealth inequalities among mothers in the low- and middle-

**Funding:** The Consortium for Advanced Research and Training in Africa (CARTA) provided logistical support to AFF in the course of writing this paper. The funders had no role in study design, data collection and analysis, decision to publish, or preparation of the manuscript. No additional external funding was received for this study.

**Competing interests:** The authors have declared that no competing interests exist.

did not have any special access that other researchers would not have.

income countries through multi-sectoral and country-specific interventions with considerations for the factors identified in this study.

## Introduction

A key target of the United Nation's Sustainable Development Goal (SDG) 3 to "ensure healthy lives and promote well-being for all at all ages" is the reduction of childhood deaths [1]. Malnutrition among under-five children is a major impediment towards the attainment of SDG 3 in Low- and Middle- Income Countries (LMICs). Combating malnutrition has remained one of the greatest global health and social challenges. Malnutrition is a prominent part of a vicious cycle that consists of both poverty and disease [2]. The trio of malnutrition, poverty and disease are interlinked. Presence or absence of one directly affects the presence or absence of the others [3]. The marginalised and vulnerable population sub-groups are the most affected. They are impoverished and also lack access to education, information, financial resources and quality healthcare. The relationship between wealth and health services uptake and health outcomes in developing countries has been established in the literature [4–9]. Fagbamigbe et al. found that persons from wealthier households in Nigeria had a higher propensity of utilizing health services [6]. However, there could be other factors associated to health outcomes and health care utilization as documented in a Ghanaian study, wherein the authors ascertained that despite free antenatal care services in Ghana, its utilization remained poor [7].

Malnutrition is one of the health outcomes with a higher level of inequality. Severe wasting is one of the health outcomes with a distinct and higher level of inequalities among millions of under-five children globally, especially in the LMICs [10]. It has been associated with myriads of interconnected factors across several individual-, household- and community-levels [11–20]. According to the literature, household food security, adequacy of health care and feeding, environmental sanitation, maternal education, parental employment status, and media access are some of the risk factors of malnutrition among children [11,12,21–23,13–20]. These factors are all connected to household wealth status.

The UNICEF framework for understanding the factors associated with malnutrition showed that economic, social, and political factors are interlinked [10]. Besides, poverty has two-edged sides to malnutrition. Poverty is a cause of malnutrition, on the one hand, and a consequence of malnutrition on the other hand [2]. Poor earnings, as a result of lack of education, joblessness or poor salary can lead to food shortages, poor sanitation and lack of health services and thereby cause malnutrition. Further, malnutrition, especially at an early age, can result in ill health and low education. Thus, malnutrition is a consequence of the factors that are closely related to one or more combinations of poor food quality, insufficient food intake, and severe and repeated infectious diseases. These conditions arise from the individual and societal standard of living, and the ability to meet necessities of life [3]. The literature is replete with the fact that malnutrition affects school absenteeism rates, cognitive development and intellectual capacity of children and thereby contribute to poor educational performances [24–26]. These outcomes can entrap individuals and societies for a long time in the cycle of poverty. An EU-WHO-TRD report on diseases of poverty otherwise referred to as the poverty-related diseases had stated that "...poverty creates conditions that favour the spread of infectious diseases and prevents affected populations from obtaining adequate access to prevention and care..." [27]. It has been reported that poor living conditions, limited access to adequate

hygienic food and potable drinking water, no medical care and lack of education promote the spread of infections.

While there are a few reports on the country-level decomposition of socioeconomic inequalities in child nutrition [8,9] with documented evidence that poverty is associated with malnutrition [28–30], we are not aware of any research that has disentangled factors associated with wealth-related inequalities in the prevalence of severe wasting (SW) among under-five children in LMICs. Whereas, the disentanglement of compositional and structural risk factors of SW by wealth inequalities would enhance the understanding of the depth and contributions of the factors associated with SW and consequently provoke evidence-based interventions. There is a need to understand how the social determinants of health can be mixed to stop or at least reduce socioeconomic inequalities in the distribution of childhood malnutrition. It is therefore pertinent to decompose the wealth-related inequalities across the risk factors associated with SW and recommend potential strategies to overcome the challenges posed by this silent child-killer. This study aims to quantify the contributions of demographic, socioeconomic and proximate factors in explaining the wealth inequality in the distribution of SW in LMICs. We hypothesised that severe wasting will be lower among children from poor households than those from non-poor households in all countries. Our study will help widen the discussion on childhood nutrition and enhance knowledge and understanding of how the social, biological and political determinants of health can be exploited to reduce socioeconomic inequalities in malnutrition. Findings from our study are potential ingredients for global and national policy and intervention in child nutrition.

## Methods

### Study design and data

The Demographic and Health Surveys (DHS) data collected periodically across the LMICs was used for this study. The DHS are cross-sectional in design and are nationally representative household surveys. We pooled data from the most recent successive DHS conducted between 2010 and 2018 and available as of March 2019 and has under-five children anthropometry data. We included only the 51 countries that met these inclusion criteria. The final data consists of 532,680 under-five children living within 55,823 neighbourhoods in 51 LMICs. In all the countries, DHS used a multi-stage, stratified sampling design with households as the sampling unit [31,32]. The DHS computes sampling weights to account for unequal selection probabilities including non-response whose application makes survey findings to fully represent the target populations. The DHS used similar protocols, standardized questionnaires, similar interviewer training, supervision, and implementation across all countries where the survey held. DHS releases different categories of data focusing on different members of households among wish we used the children recode data for the current study. The data covered the birth history and health experiences of under-five children born to sampled women within five years preceding the survey date. The anthropometry measurements were taken using standard procedures [33,34]. The full details of the sampling methodologies are available at dhsprogram.com.

**Dependent variable.** The outcome variable in this study is severe wasting. It is defined as "the presence of muscle wasting in the gluteal region, loss of subcutaneous fat, or prominence of bony structures, particularly over the thorax" [35] and approximated by "a very low weight for height score (WHZ) below -3 z-scores of the median WHO growth standards, by visible severe wasting, or by the presence of nutritional oedema" [12] more so, malnutrition has been recently described as "related to both deficiencies and excesses in nutrition, and then, therefore, it includes wasting, stunting, underweight, micronutrient deficiencies or excesses,

overweight, and obesity" [36]. SW was a composite score of children weight and height. We generated z-scores using WHO-approved methodologies [37] and categorized children with z-scores <-3 standard deviation as having SW (Yes = 1), otherwise as No = 0.

**Main determinant variable.** In this decomposition study, household wealth status computed as a composite score of assets owned by households was used as a proxy for family income as DHS does not collect data on family earnings or expenditures. The methods used in computing DHS wealth index have been described previously [38]. Additional details of the methods and assets used for the computation of the wealth quintiles is available at dhsprogram.com. The DHS data had already generated and categorized household wealth quintile as a variable into 5 categories of 20% each: poorest, poorer, middle, richer and richest. For the decomposition analysis, we re-categorized household wealth quintile into two categories: poor (poorest, poorer) and non-poor (middle, richer and richest). A similar categorization has been used elsewhere [8,9,39,40]. Hence, we define "wealth inequality" as "the unequal distribution of assets".

**Independent variables.** Keywords including low and middle-income countries, childhood morbidity, undernutrition, malnutrition, severe acute malnutrition, severe wasting, were used to search for factors associated with wealth-based inequality in SW across literature database such as PubMed, Medline, Hinari. The individual- and neighbourhood level factors were identified empirically from the literature [11–23,41] are:

**Individual-level factors.** The individual-level factors are the sex of the children (male versus female): to determine if the biological differences could explain susceptibility to SW; children age in years (under 1 year and 12–59 months): SW has been reported to differ by children ages; maternal education (none, primary or secondary plus): better education could lead to better access to information and enhance earnings, and reduced risk of SW; maternal age (15 to 24, 25 to 34, 35 to 49): younger mothers may have limited education and earnings and thereby increase risk of SW among their children. Others are marital status (never, currently and formerly married): currently married may have spousal support that may reduce the risk of SW; occupation (currently employed or not): capability of providing necessary nutritional intakes; access to media (at least one of radio, television or newspaper): access to information could enhance prevention of SW; sources of drinking water (improved or unimproved), toilet type (improved or unimproved), weight at birth (average+, small and very small), birth interval (firstborn, <36 months and >36 months): children with short birth interval are at higher risk of SW and may have higher experience of wealth-related inequality in SW; and birth order (1, 2, 3 and 4+), children with high birth order are at higher risk of SW and experience higher wealth-related inequality in SW [11–23,41].

**Neighbourhood-level factors.** We used the word "neighbourhood" to describe the clustering of the children within the same geographical environment. Neighbourhoods were based on sharing a common primary sample unit (PSU) within the DHS data [31,32]. Operationally, we defined "neighbourhood" as clusters and "neighbours" as members of the same cluster. The PSUs were identified using the most recent census in each country where DHS was conducted. We considered neighbourhood socioeconomic disadvantage as a neighbourhood-level variable in this study. Neighbourhood socioeconomic disadvantage was operationalized with a principal component comprised of the proportion of respondents without education (poor), unemployed, living in rural areas, and living below the poverty level [11–23,41].

## Statistical analyses

In this study, we carried out descriptive statistics and analytical analyses comprising of bivariable analysis and Blinder-Oaxaca decomposition techniques using binary logistic regressions.

Descriptive statistics was used to show the distribution of respondents by country and key variables. Estimates were expressed as percentages alongside 95% confidence intervals. We computed the risk difference in the development of SW between under-five children from poor and non-poor households. A risk difference (RD) greater than 0 suggests that SW are prevalent among children from poor households (pro-poor inequality). A negative RD indicates that SW is prevalent among children from non-poor households (pro-non-poor inequality). We estimated the fixed effects as the weighted risk differences for each of the country and the random effect as the overall risk difference irrespective of a child's country of residence.

Lastly, the logistic regression method was applied to the pooled cross-sectional data from the 51 LMICs to carry out a Blinder-Oaxaca decomposition analysis (BODA). The BODA is an approach to examine differences in outcomes between groups is the decomposition technique developed by Oaxaca and Blinder [42,43]. This method aims to explain how much of the difference in mean outcomes across two groups is due to group differences in the levels of the independent variables, and how much the difference can be attributed to the differences in the magnitude of regression coefficients [42,43].

The method decomposes the differences in an outcome variable between 2 groups into 2 components so that the gaps between the two groups can be more visible. The first component of the decomposition is the "explained" portion of the gap that captures differences in the distributions of the measurable characteristics (also known as the "compositional" or "endowments") of these groups. The endowment effect captures differences in the outcome of interest that arises from observed differentials in the characteristics between the groups. Also, the second components of the analysis called the structural or coefficient or return effect, is unexplained and is attributed to differences in the returns to endowments between groups. Thus, each group receives different returns for the same level of endowments. In the analysis of health outcomes, the effect of the return may reflect the indirect effects of structural differences in health systems that affect the healthcare utilization between different groups. In recent time, the classical BODA has been extended from continuous outcomes to binary and other non-linear outcomes [40–43].

We, therefore, adopted this technique to enable the quantification of how much of the gap between the "advantaged" (non-poor) and the "disadvantaged" (poor) groups is attributable to differences in specific measurable characteristics. The non-linear decomposition model assumes that the conditional expectation of the probability of a child having SW is a non-linear function of a vector of characteristics. Using the generalized structure of the model, we fitted a model each for children born to poor and non-poor mothers.

**The methodologies of Blinder Oaxaca Decomposition Analysis (BODA).** The BODA is a statistical method that decomposes the gap in the mean outcomes across two groups into a portion that is due to differences in group characteristics and a portion that cannot be explained by such differences. Therefore, Let A and B be two group names for children from households in poor and non-poor wealth quintiles. Also, let $\bar{Y}_A$ and $\bar{Y}_B$ be the mean outcomes for the observations Y in the groups so that the mean outcome difference ($Ð\bar{Y}$) to be explained is the difference between $\bar{Y}_A$ and $\bar{Y}_B$.

Then the mean outcome for group G can be written as:

$$Y_\ell = X'_\ell \beta_\ell + \epsilon_\ell, \ E(\epsilon_\ell) = 0, \ \ell \in \{A, B\} \tag{1}$$

where X is a vector containing the predictors and a constant, $\beta$ contains the slope parameters and the intercept, and $\epsilon$ is the error, the mean outcome difference can be expressed as the difference in the linear prediction at the group-specific means of the regressors. That is:

$$Ð\bar{Y} \ = \ \bar{Y}_A - \bar{Y}_B \ = \ E(X_A)'\beta_A - E(X_B)'\beta_B \tag{2}$$

Since

$$\mathrm{E}(Y_\ell) = \mathrm{E}(X'_\ell \beta_\ell + \epsilon_\ell) = \mathrm{E}(X'_\ell \beta_\ell) + E(\epsilon_\ell) = \mathrm{E}(\mathrm{X}_\ell)' \beta_\ell$$

assuming that $E(\beta_\ell) = \beta_\ell$ and $E(\epsilon_\ell = 0)$.

Then the contribution of group differences in predictors to the overall outcome difference can be identified by rearranging Eq 2 to give:

$$Ð\bar{Y} = \{E(\mathrm{X}_A) - \mathrm{E}(\mathrm{X}_B)\}' \beta_B + \mathrm{E}(\mathrm{X}_B)'(\beta_A - \beta_B) + \{E(\mathrm{X}_A) - \mathrm{E}(\mathrm{X}_B)\}'(\beta_A - \beta_B) \qquad (3)$$

In Eq (3), we have divided the outcome difference into three parts thus $Ð\bar{Y} = E + C + I$ in the viewpoint of group B so that the group differences in the predictors are weighted by the coefficients of group B to determine the endowment effects. Where E = $E(\mathrm{X}_A) - \mathrm{E}(\mathrm{X}_B)\}'\beta_B$; is the part of the differentials due to group differences in the predictors that is the "endowment effect", C = $\mathrm{E}(\mathrm{X}_B)'(\beta_A - \beta_B)$, is the measure of the contribution of differences in the coefficients which includes the differences in the intercept and lastly, $I = \{E(\mathrm{X}_A) - E(\mathrm{X}_B)\}'(\beta_A - \beta_B)$is the measure of the interaction term accounting for the fact that differences in endowments and coefficients exist simultaneously between the two groups. The *E* component measures the expected change in group *B*'s mean outcome if group *B* had group *A*'s predictor levels. Similarly, for the *C* component (the "coefficients effect"), the differences in coefficients are weighted by group *B*'s predictor levels. That is, the *C* component measures the expected change in group *B*'s mean outcome if group *B* had group *A*'s coefficients [42,44,45].

In this study, we adopted an alternative (further) decomposition from the concept that there is a nondiscriminatory coefficient vector that should be used to determine the contribution of the differences in the predictors. We assumed $\beta^*$ to be a nondiscriminatory coefficient vector. The outcome difference can then be written as:

$$Ð\bar{Y} = \{E(\mathrm{X}_A) - \mathrm{E}(\mathrm{X}_B)\}'\beta^* + \{\mathrm{E}(\mathrm{X}_A)'(\beta_A - \beta^*) + \mathrm{E}(\mathrm{X}_B)\}'(\beta^* - \beta_B)\} \qquad (4)$$

which can be thought of as $Ð\bar{Y} = Q + \mathrm{U}$ wherein $Q = E(\mathrm{X}_A) - E(\mathrm{X}_B)\}'\beta^*$ is the part of the outcome differential that is explained by group differences in the predictors (the "quantity effect"), and the second component, U = $E(\mathrm{X}_A)'(\beta_A - \beta^*) + E(\mathrm{X}_B)'(\beta^* - \beta_B)$is the "unexplained" part. This part is attributed to discrimination, and also captures all the potential effects of differences in unobserved variables.

The unknown nondiscriminatory coefficients vector $\beta^*$ can be estimated thereafter by assuming that $\beta^* = \beta_A$ or $\beta^* = \beta_B$ [42], wherein discrimination is directed against A and none against group B, then $\_\beta_A$ can be used as an estimate for $\beta^*$ as:

$$Ð\bar{Y} = (\bar{\mathrm{X}}_A - \bar{\mathrm{X}}_B)'^{\hat{\beta}}{}_A + \bar{\mathrm{X}}'_A(\hat{\beta}_A - \hat{\beta}_B)' \qquad (5)$$

and vice-versa. The numerical details have been reported [44,45].

The DHS stratification and the unequal sampling weights of clusters, as well as household clustering effects, were considered. Hence we weighted the data and set significance to 5%. Data were analysed using R statistical software and STATA 16 (StataCorp, College Station, Texas, United States of America).

The results of this study are presented in Tables and Figures. All our estimates were weighted. In Table 1, we present the proportion of children from households in the poor wealth quintiles and the prevalence of SW by countries. Also, we present the prevalence of SW among the children from households in the poor and non-poor wealth quintiles within each country. The distribution of the children by the characteristics studied the prevalence of SW by the levels of the characteristics and result of the test of association between the characteristics and the development of SW.

**Table 1. Distribution of the children by countries, poverty and prevalence of severe wasting among under-five children in LMICs, DHS 2010–2018.**

| Country | Year of Survey | Number of Under-5 Children | Weighted SW prevalence (%) | Weighted Poor (%) | *Weighted SW (%) Poor | Weighted SW (%) Non-poor |
|---|---|---|---|---|---|---|
| All | | 532,680 | 4.7 | 45.6 | 5.3 | 4.2 |
| Eastern Africa | | 67,418 | 1.5 | 45.6 | 2.0 | 1.2 |
| Burundi | 2016 | 6,052 | 0.9 | 42.5 | *1.4 | 0.5 |
| Comoro | 2012 | 2,387 | 3.9 | 47.1 | 4.6 | 3.2 |
| Ethiopia | 2016 | 8,919 | 3.0 | 46.8 | *3.5 | 2.6 |
| Kenya | 2014 | 18,656 | 1.0 | 45.2 | *1.3 | 0.7 |
| Malawi | 2016 | 5,178 | 0.6 | 47.5 | 0.5 | 0.7 |
| Mozambique | 2011 | 9,313 | 2.1 | 45.6 | *2.9 | 1.4 |
| Rwanda | 2015 | 3,538 | 0.6 | 46.8 | 0.7 | 0.6 |
| Tanzania | 2016 | 8,962 | 1.3 | 46.4 | 1.5 | 1.0 |
| Uganda | 2016 | 4,413 | 1.4 | 43.2 | *1.9 | 1.0 |
| Middle Africa | | 37,136 | 2.5 | 44.4 | 2.7 | 2.3 |
| Angola | 2016 | 6,407 | 1.0 | 45.4 | *1.5 | 0.7 |
| Cameroon | 2010 | 5,033 | 1.9 | 44.3 | *3.1 | 0.8 |
| Chad | 2015 | 9,826 | 4.3 | 42.5 | 3.8 | 4.6 |
| Congo | 2012 | 4,475 | 1.6 | 47.8 | 2.1 | 1.1 |
| DRC | 2014 | 8,059 | 2.7 | 45.1 | *3.1 | 2.3 |
| Gabon | 2012 | 3,336 | 1.2 | 43.1 | 0.9 | 1.3 |
| Northern Africa | | 13,682 | 3.8 | 37.5 | 3.4 | 4.0 |
| Egypt | 2014 | 13,682 | 3.8 | 37.5 | 3.4 | 4.0 |
| Southern Africa | | 20,273 | 1.7 | 46.5 | 2.0 | 1.4 |
| Lesotho | 2016 | 1,312 | 0.7 | 42.3 | *1.4 | 0.2 |
| Namibia | 2013 | 1,558 | 2.2 | 47.3 | *3.1 | 1.3 |
| South Africa | 2016 | 1,082 | 0.5 | 47.5 | 0.6 | 0.4 |
| Zambia | 2014 | 11,407 | 2.1 | 47.7 | 2.3 | 1.9 |
| Zimbabwe | 2015 | 4,914 | 1.1 | 44.4 | *1.5 | 0.8 |
| Western Africa | | 85,462 | 4.7 | 44.0 | 5.4 | 4.2 |
| Benin | 2018 | 12,033 | 1.1 | 41.6 | 1.1 | 1.0 |
| Burkina Faso | 2010 | 6,532 | 5.8 | 42.0 | 6.4 | 5.5 |
| Cote d'Ivoire | 2012 | 3,200 | 1.8 | 49.4 | 1.9 | 1.8 |
| Gambia | 2013 | 3,098 | 4.7 | 46.0 | 4.2 | 5.1 |
| Ghana | 2014 | 2,720 | 0.7 | 43.2 | *1.1 | 0.4 |
| Guinea | 2012 | 3,085 | 3.7 | 46.1 | 4.1 | 3.4 |
| Liberia | 2013 | 3,171 | 2.2 | 47.8 | 2.5 | 1.9 |
| Mali | 2013 | 4,306 | 5.1 | 42.5 | *6.2 | 4.2 |
| Niger | 2012 | 4,771 | 6.2 | 39.5 | *6.7 | 5.8 |
| Nigeria | 2013 | 24,505 | 8.8 | 43.9 | *10.6 | 7.5 |
| Senegal | 2017 | 10,787 | 1.5 | 46.8 | *2.1 | 1.0 |
| Sierra Leone | 2013 | 4,069 | 3.8 | 47.1 | 3.9 | 3.7 |
| Togo | 2014 | 3,185 | 1.6 | 43.1 | 1.7 | 1.5 |
| Central Asia | | 9,883 | 1.5 | 39.4 | 1.3 | 1.7 |
| Kyrgyz | 2012 | 4,016 | 1.1 | 39.2 | 1.2 | 1.0 |
| Tajikistan | 2017 | 5,867 | 1.8 | 39.4 | 1.4 | 2.1 |
| South-Eastern Asia | | 9915 | 6.6 | 44.6 | 7.3 | 6.0 |
| Myanmar | 2016 | 4,197 | 1.4 | 52.1 | 1.4 | 1.4 |
| Timor-Leste | 2016 | 5,718 | 9.9 | 39.9 | *12.3 | 8.4 |

*(Continued)*

**Table 1.** (Continued)

| Country | Year of Survey | Number of Under-5 Children | Weighted SW prevalence (%) | Weighted Poor (%) | *Weighted SW (%) Poor | Weighted SW (%) Non-poor |
|---|---|---|---|---|---|---|
| Southern Asia | | 245,173 | 7.0 | 46.8 | 7.8 | 6.4 |
| Bangladesh | 2014 | 6,965 | 3.1 | 41.5 | *3.6 | 2.7 |
| India | 2016 | 225,002 | 7.4 | 47.2 | *8.2 | 6.8 |
| Maldives | 2016 | 2,362 | 2.0 | 44.7 | 2.0 | 1.9 |
| Nepal | 2016 | 2,369 | 1.9 | 42.2 | 2.1 | 1.7 |
| Pakistan | 2018 | 4,151 | 2.3 | 42.0 | *3.3 | 1.6 |
| Cambodia | 2014 | 4,324 | 2.4 | 44.4 | 2.6 | 2.3 |
| Western Asia | | 1561 | 1.5 | 40.4 | 1.9 | 1.2 |
| Armenia | 2016 | 1561 | 1.5 | 40.4 | 1.9 | 1.2 |
| Central America | | 21,717 | 0.2 | 47.6 | 0.2 | 0.2 |
| Guatemala | 2012 | 11,744 | 0.1 | 48.8 | 0.1 | 0.1 |
| Honduras | 2016 | 9,973 | 0.3 | 45.9 | 0.3 | 0.2 |
| South America | | 9,213 | 0.1 | 47.5 | 0.1 | 0.1 |
| Peru | 2012 | 9,213 | 0.1 | 47.5 | 0.1 | 0.1 |
| South Europe | | 2,462 | 0.5 | 44.5 | 4.3 | 0.3 |
| Albania | 2018 | 2,462 | 0.5 | 44.5 | 0.7 | 0.3 |
| Caribbean | | 8795 | 0.8 | 46.3 | 0.9 | 0.6 |
| Dominica | 2013 | 3,187 | 0.6 | 45.6 | 0.4 | 0.6 |
| Haiti | 2016 | 5,598 | 0.9 | 46.6 | 1.2 | 0.6 |

*Significant at 0.05 in Mantel Haenszel test of homogeneity of the odds ratio.

**Ethics approval and consent to participate.** This study was based on an analysis of existing survey data with all identifier information removed. The survey was approved by the Ethics Committee of the ICF Macro at Fairfax, Virginia in the USA and by the National Ethics Committees in their respective countries. All study participants gave informed consent before participation and all information was collected confidentially. The full details can found at http://dhsprogram.com.

## Results

### Sample characteristics

In Table 1, we listed the year of the survey, the numbers of neighbourhoods where data was collected, the population of under-five children surveyed, the weighted prevalence of SW, percentage of children from poor households, and the prevalence of SW among children from poor and non-poor households by countries and the regions of the world. The proportion of children from poor households ranged from 37.5% in Egypt to 52.1% in Myanmar. The overall SW prevalence was 4.7% while the overall poor and non-poor dichotomy in SW prevalence was 5.3% versus 4.2%, with statistically significant differences as shown in Table 1 and Fig 1. The prevalence of SW among children from poor households ranged from 0.1% in Guatemala to 12.3% in Timor-Leste, while it ranged from 0.1% in Guatemala to 8.4% in Timor-Leste among children from non-poor households.

Table 2 presents the descriptive statistics for the pooled sample of children across the 51 LMICs by their sociodemographic and reproductive characteristics. About 51% of the children were male while only 20% were infants. About 53% were from mothers aged 25 to 34 years old and about 41% had no formal education. Nearly one-third of the mothers were not working at

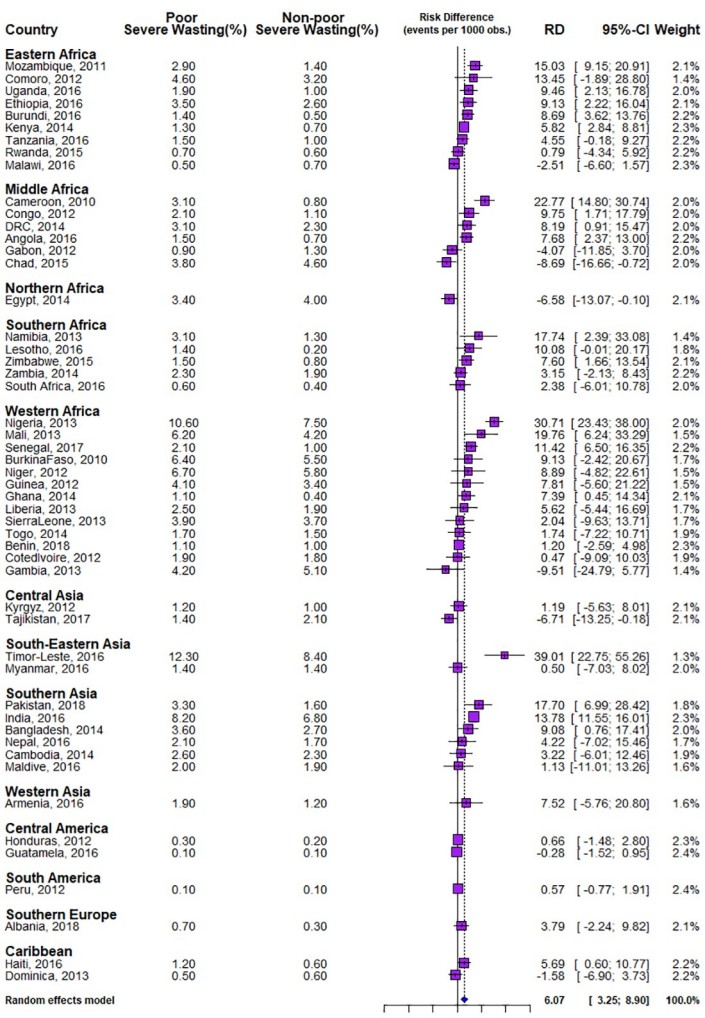

**Fig 1. Risk difference in the prevalence of severe wasting between children from poor and non-poor households by countries.**

the time of the survey. The overall prevalence of SW in the group of children from poor households was 5.3% compared with 4.2% among those from non-poor households. Prevalence of SW was consistently higher among children from poor households compared with those from non-poor households across all the background characteristics considered in this study.

**Magnitude and variations in poverty inequality in severe wasting.** In Figs 1 and 2, we showed the risk difference of the level of inequality between children from poor and non-poor households across the 51 LMICs included in this study. Of the 51 countries, 21 countries showed statistically significant pro-poor inequality (i.e. SW was more prevalent among children from poor households). Only three countries showed statistically significant pro-non-poor inequality (i.e. SW was prevalent among children from non-poor households) while 27 countries showed no statistically significant inequality. As illustrated in Fig 1, in Eastern Africa, the educational difference was largest for Mozambique (15.03 per 1000 children) and lowest for Malawi (−2.51). In Middle Africa, the largest risk difference was found in Cameroun (22.77) and least in Chad (-8.69). In Western Africa, the largest pro-poor difference was in Nigeria (30.71) and lowest for Gambia (-9.51). In South-Eastern Asia, the difference was

**Table 2. Summary of pooled sample characteristics of the studied children in 51 LMICs.**

| Characteristics | Weighted n | Weighted % | Weighted (%) Poor (%) | Weighted SW (%) Poor | Weighted SW (%) non-poor |
|---|---|---|---|---|---|
| Individual Level | 532,680 | | 45.6 | 5.3 | 4.2 |
| Age | | | | | |
| <12 Months | 103,379 | 20.0 | 45.3 | 8.1 | 6.8 |
| 12–59 Months | 413,718 | 80.0 | 45.7 | 4.5 | 3.5 |
| Sex | | | | | |
| Female | 252,541 | 48.8 | 46.1 | 4.8 | 3.8 |
| Male | 264,556 | 51.2 | 45.1 | 5.7 | 4.6 |
| Maternal Age | | | | | |
| 15–24 | 160,133 | 31.0 | 47.1 | 5.7 | 4.8 |
| 25–34 | 273,802 | 52.9 | 43.4 | 5.2 | 4.2 |
| 35–49 | 83,162 | 16.1 | 49.8 | 4.5 | 3.1 |
| Maternal Education | | | | | |
| None | 165,629 | 31.1 | 67.6 | 6.3 | 4.7 |
| Primary | 134,578 | 25.3 | 53.2 | 3.4 | 2.7 |
| Secondary+ | 231,738 | 43.6 | 25.4 | 5.4 | 4.6 |
| Employment | | | | | |
| Yes | 366,033 | 70.8 | 46.4 | 5.6 | 4.4 |
| No | 151,064 | 29.2 | 43.5 | 4.3 | 3.6 |
| Access to Media | | | | | |
| No | 188,357 | 36.5 | 70.9 | 5.8 | 4.0 |
| Yes | 328,311 | 63.5 | 31.1 | 4.5 | 4.2 |
| Drinking-Water Sources | | | | | |
| Unimproved | 95,544 | 19.2 | 66.1 | 4.5 | 3.3 |
| Improved | 402,688 | 80.8 | 40.9 | 5.5 | 4.3 |
| Toilet Type | | | | | |
| Unimproved | 248,331 | 49.9 | 68.7 | 5.6 | 4.1 |
| Improved | 249,753 | 50.1 | 22.9 | 4.3 | 4.1 |
| Marital Status | | | | | |
| Never Married | 12,199 | 2.4 | 37.3 | 2.3 | 1.6 |
| Currently Married | 484,949 | 93.8 | 45.7 | 5.4 | 4.4 |
| Formerly Married | 19,946 | 3.9 | 47.8 | 2.9 | 1.9 |
| Weight At Birth | | | | | |
| Average+ | 423,017 | 85.4 | 44.5 | 5.1 | 4.2 |
| Small | 52,939 | 10.7 | 49.5 | 5.8 | 4.2 |
| Very Small | 19,624 | 4.0 | 52.3 | 7.1 | 5.6 |
| Birth Interval | | | | | |
| 1st | 157,067 | 30.4 | 37.5 | 5.5 | 4.5 |
| <36 | 193,030 | 37.4 | 52.8 | 5.4 | 4.4 |
| 36+ | 165,780 | 32.1 | 45.0 | 4.9 | 3.6 |
| Birth Order | | | | | |
| 1 | 157,065 | 30.4 | 37.5 | 5.5 | 4.5 |
| 2 | 134,436 | 26.0 | 40.8 | 5.3 | 4.6 |
| 3 | 83,134 | 16.1 | 48.4 | 5.5 | 3.8 |
| 4 | 142,462 | 27.6 | 57.5 | 5.0 | 3.5 |
| Neighbourhood Factors | | | | | |
| Residence | | | | | |
| Rural | 368,461 | 69.3 | 59.8 | 5.3 | 4.3 |

*(Continued)*

**Table 2.**  (Continued)

| Characteristics | Weighted n | Weighted % | Weighted (%) Poor (%) | Weighted SW (%) Poor | Weighted SW (%) non-poor |
|---|---|---|---|---|---|
| Urban | 163,510 | 30.7 | 13.6 | 4.8 | 4.1 |
| Community SES Quintiles | | | | | |
| 1 (Highest) | 117,186 | 20.2 | 9.1 | 4.6 | 4.2 |
| 2 | 101,302 | 20.0 | 24.9 | 4.8 | 4.2 |
| 3 | 103,795 | 20.1 | 45.8 | 4.6 | 3.9 |
| 4 | 100,611 | 20.0 | 69.0 | 5.2 | 4.4 |
| 5 (Lowest) | 94,203 | 19.7 | 88.1 | 5.9 | 4.9 |
| Total | 532,680 | 100.0 | 45.6 | 5.3 | 4.2 |

largest for Timor-Leste (39.01) and lowest for Dominica (-1.58). The largest difference in Southern Asia was found in Pakistan (17.70) compared with the lowest (1.13) found in the Maldives. In the pooled analysis, irrespective of region, Timor-Leste had the highest pro-poor inequality (39.01), followed by Nigeria (30.71) and Cameroun (22.77) and least in Chad (-8.69) as shown in Figs 1 and 2. Overall, there was significant pro-poor in the total pooled sample of children in this study. The random effect model showed that the overall risk difference was 6.07 (95% CI: 2.8–9.6) per 1000 children among children from poor households compared with those from non-poor households as shown in Fig 1.

Statistically significant pro-poor inequality was found in five of the nine countries in Eastern Africa, 3 of the 6 countries in Middle Africa, two countries in Southern Africa. In Western Africa, 3 of the 13 countries showed statistically significant pro-poor inequality, 3 countries in Southern Asia and the two countries studied in South-Eastern Asia. Also, statistically significant pro-non-poor inequality was found in Chad in Western Africa, Egypt in the Northern African region, and Tajikistan in Central Asia.

**Relationship between prevalence of severe wasting and magnitude of poverty inequality.** Fig 3 shows the relationship between the prevalence of SW and the magnitude of inequality for each of the 51 countries in this study. We categorized the 51 countries into 4 distinct categories based on the level of SW (low/high) and level of pro-poor inequality.

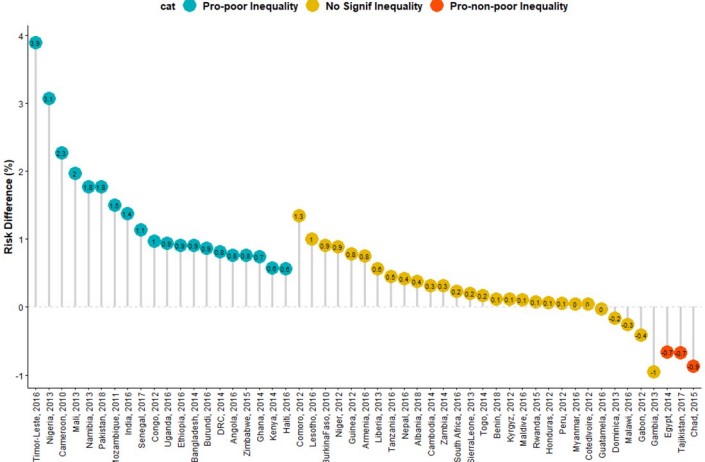

**Fig 2. Risk difference in having severe wasting between children from poor and non-poor households by countries.**

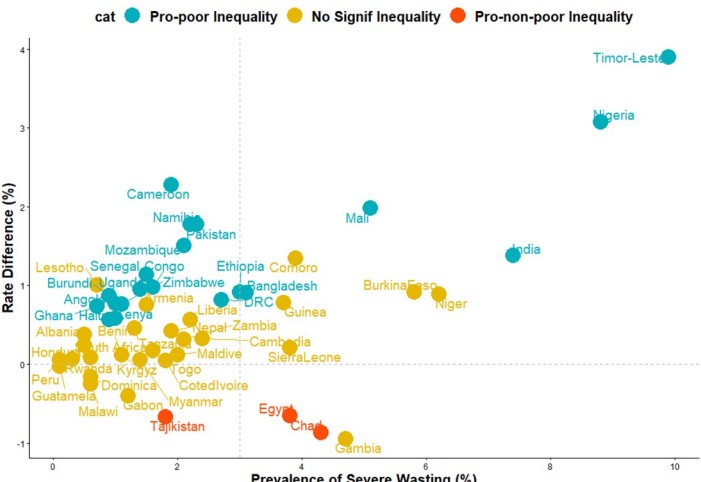

**Fig 3. Scatter plot of prevalence of severe wasting and risk difference between children from poor and non-poor households in LMICs.**

1. High severe wasting and high pro-poor inequality such as Nigeria and Timor-Leste.

2. High severe wasting and high pro-non-poor inequality such as Chad and Egypt.

3. Low severe wasting and high pro-poor inequality such as Uganda and Namibia.

4. Low severe wasting and high pro-non-poor inequality such as Tajikistan.

**Decomposition of socioeconomic inequality in the prevalence of severe wasting.** In Fig 4, we showed the detailed decomposition of the part of the inequality that was caused by compositional effects of the determinants of SW among under-five children. Only 20 countries were identified to have statistically significant differences viz-a-viz the distribution of SW by pro-poor inequalities. Across the countries, there were variations in the effect of the factors associated with wealth inequalities. For the full details of the decomposition analysis, see S1 Table. In Fig 4, the values in the boxes represent the percentage gap (differences between the compositional 'explained' components and the structural 'unexplained' components) in the influence of the variables on poor-non-poor gaps across each country. The positive values in the boxes signify that the compositional 'explained' components exceeded the structural 'unexplained' components while the negative values show the reverse. For instance, the -871% for neighbourhood social-economic disadvantage in Lesotho showed that there was wide variation in the contribution of neighbourhood social-economic indicators to the distribution of having SW in Lesotho viz-a-viz the unexplained components in the poor-non-poor inequalities in SW.

On average, neighbourhood socioeconomic status disadvantage and location of residence were the most important factors in most countries. In Senegal, the largest contributors to the socioeconomic inequality in the prevalence of SW as neighbourhood socioeconomic disadvantage, followed by the location of residence, maternal age and access to media. Maternal age and media access narrowed the inequality in the development of SW between children from non-poor and poor mothers in most countries. In India, birth interval and birth order contributed mostly to SW. In Namibia, maternal age, birth weight and access to media contributed mostly to SW. The sex and age of the child, marital status and source of drinking water did not

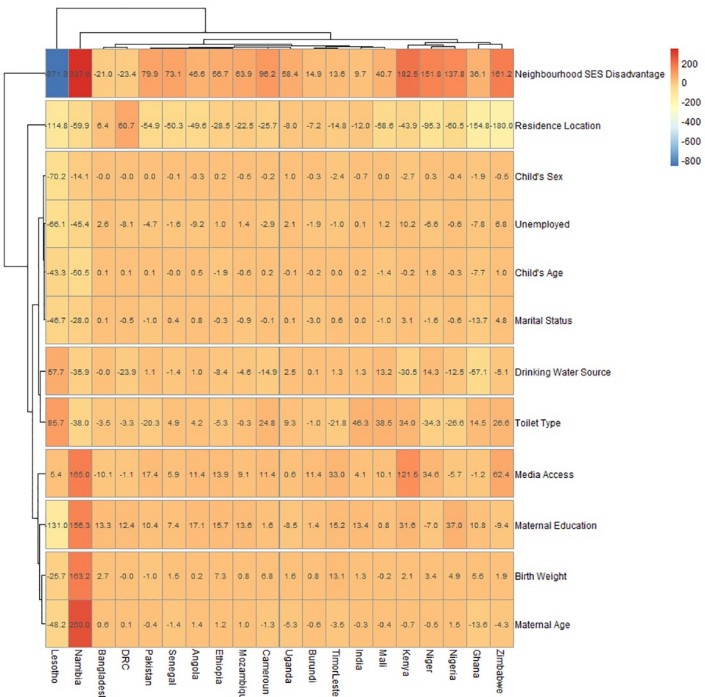

**Fig 4. Contributions of differences in the distribution 'compositional effect' of the determinants of SW to the total gap between children from poor and non-poor mothers by countries.**

show any significant contribution to socioeconomic inequality in the development of SW in any of the 20 countries identified to have significant compositional differences. The highest contributors to the inequality in Timor-Leste are toilet types, neighbourhood socioeconomic status, media access, maternal education and place of residence.

## Discussion

Severe wasting is currently affecting millions of children across most LMICs and the burden persisted despite the attention it has attracted over the years. The protracted and precarious nutritional outcome among under-five children motivated this study. Using pooled data from DHS in 51 LMICs, we identified the pattern of SW among under-five children, its and the contextual and compositional factors associated with its socioeconomic inequality. In all, our findings showed that children from non-poor households had a lower likelihood of SW. This is consistent with previous reports [8,9,46]. We found wide variations in the prevalence of SW among the children from poor and non-poor women across the studied countries. The prevalence of SW among the children from poor and non-poor households ranged from 0.1% in Guatemala to 12.3% in Timor-Leste and from 0.1% in Guatemala to 8.4% in Timor-Leste respectively. It is worth noting that about 53% of their mothers were of active childbearing age (25–34 years) and nearly a third had no formal education and about 30% were employed as of the survey time whereas two-thirds reside in the rural areas. Each of these factors propels poor economic capabilities. Besides, we found a higher prevalence of SW among children from neighbourhoods with the highest socioeconomic disadvantage irrespective of whether the children are from poor households or not.

Our analysis revealed significant and wide differentials in the poor and non-poor gap across various determinants of SW. Our finding is collaborated by earlier studies that reported education, age, media access, birth weight, child sex and place of residence among others as associated with SW [15,16,25,29,30,47–50]. These factors provided a plausible explanation of the variations in the prevalence of SW among the children from poor and non-poor households. The prevalence of SW was consistently higher among children from poor households compared with those from non-households across all the background characteristics considered in this study. We also found disparities in the prevalence of SW by sex and age with the infants and male children at higher risk of SW.

We found good evidence of inter-country differences in the risk-difference in the distribution of SW between the children from poor and non-poor households. The analysis of risk difference of SW between the children from poor and non-poor households in each country revealed the rather obscured variations in the differences. The largest disparities were in Nigeria where a difference of 30 children among 1000 who have SW were from poor households. Overall we found a risk difference of 6 children per every 1000 children to have SW between children from poor and non-poor households. This finding suggests a relationship between poverty and SW. Children from poor households have a higher likelihood of developing SW than children from non-poor households. In general, older mothers, higher maternal education, access to media, improved sources of drinking water and toilet types are associated with a lower risk of SW. Also, children with at least an "average" low birth weight, with over 3 years preceding birth intervals, and higher birth orders had a lower risk of SW.

In the majority of the countries, the prevalence of SW was higher among the children from poor households than among those from non-poor households, with exceptions of pro-non-poor countries (Egypt, Chad and Tajikistan). We had hypothesised that children nutritional outcomes would be better among children from poor households than those from non-poor households. However, our findings proved otherwise in 3 of the countries. This finding is of important concern. Literature check showed that Chad failed in her drive to achieve the millennium development goals on malnutrition [51]. This was partly attributable to barriers to optimal feeding practices [52]. Also, Chad ranked one of the least on the Global Hunger Index (the combination of wasting, stunting, undernourishment, and under-five mortality) [52,53]. Besides, Mcnamara et al. had noted that "interactions between food security and local knowledge negotiated along multiple axes of power" including political and economic systems, health beliefs and food taboos which influence household nutrition in Chad [54]. For Tajikistan, a country with the largest share of remittances to GDP in the world has very slow progress in halting its high levels of child malnutrition [55]. Coupled with migration [55], the country has been unable to match her vast poverty reduction from 83% in 2000 to 30% in 2016 [56] and with a projected fall to 26% by 2019 [57] to a significant reduction in child malnutrition. In Egypt, inadequate dietary intake as a result of poor infant and young child feeding practices birthed the reported consistent decline in exclusive breastfeeding from 34% in 2005 to 13% in 2014, food insecurity, unbalanced diet, and "poor dietary habits, lifestyle and lack of nutritional awareness across the population, as opposed to issues of food availability" [58] as well as poor environmental conditions with only a third having improved toilet types [58]. These factors might have put the children from non-poor households at higher risk of severe wasting in the 3 countries.

Pro-poor inequality was more prominent in Eastern, Middle, Southern, Western Africa, Southern Asia and in the Caribbean than in other regions. The overall pro-poor inequalities across the studied children is a pointer that due attention has not been paid to wealth inequalities in child nutrition across the world. Therefore, there is a need to design malnutrition intervention(s) programmes with a focus on wealth-related inequalities if the problem of SW

worldwide is to be tackled successfully. The countries that showed low yet significant pro-poor inequality were Cameroun, Lesotho, Ghana, Burundi, Haiti, Kenya, Zimbabwe, Uganda, Senegal, DRC and Mozambique while countries such as Pakistan, Ethiopia, Bangladesh, Mali, Niger, India, Nigeria, and Timor-Leste had high SW and high pro-poor inequality. Also, Tajikistan had low but pro-non-poor inequality whereas Chad and Egypt had high SW prevalence and high pro-non-poor inequality. It may be necessary for these countries to learn what works and what does not work in other countries that do not have high wealth inequalities to attain the SDG on health for all. It is striking that SW is more likely among children born to currently married and employed women as of survey time.

The decomposition analysis to understand the factors that contribute to poverty inequality in the prevalence of SW by countries and to identify the relative gap between poor and non-poor households showed that the contributions of the compositional 'explained' and structural 'unexplained' components varied across countries. Previous studies reported that malnutrition does not necessarily affect growth inequality in under-five children in some countries [46]. This is a pointer that other compositional effects contribute to SW inequalities. Compositional effects, majorly from neighbourhood socioeconomic status (SES) disadvantage, birth interval, birth order, Media access, maternal education, birth weight and maternal age were responsible for most of the inequality in SW between the children from poor and non-poor households. These compositional factors were most noticeable in Lesotho, Namibia, Kenya, Zimbabwe, Cameroun, Niger, Nigeria and India. However, in Lesotho and India, the structural effects were attributable to most of the socioeconomic inequality in SW between the children of poor and non-poor households. In India, birth interval and birth order were the major effects and they contributed to the compositional and structural components respectively in the country.

In our analysis, Timor-Leste is an outlier at both the prevalence of severe wasting and in the decomposition analysis. Our finding is in tandem with earlier reports that Timor-Leste's under-five wasting prevalence was 11%, higher than 9% average in the developing countries [59]. This could be ascribed to the country's poor nutritional intakes as only 50% of infants had exclusive breastfeeding and a high burden of malnutrition among its adult population [59]. The decomposition analysis showed that the greatest contributors to pro-poor inequalities in severe wasting in Timor-Leste were poor media access, low birth weight, low maternal education, unimproved toilet type, residing in rural areas and neighbourhood socioeconomic disadvantage. Implementing necessary interventions with focus on the highlighted factors will help bridge the socioeconomic inequality gap and also reduce the prevalence of severe wasting in Timor-Leste.

Neighbourhood SES disadvantage was associated with a high prevalence of SW in all the countries. Other major contributors to the inequality effects are media access, maternal age and parental education. This is consistent with reports from local, national and international studies on the effect of socio-economic status on nutritional outcomes among under-five children [46,60–63]. The role of the media in nutrition cannot be over-emphasized. Access to media through television, radio or newspaper is very vital to avail the mothers the up-to-date information that can be useful in enhancing child nutrition. Access to media reflects the increasing recognition that there is a web of factors that influence health interventions including child nutrition. A child whose mother has better education, exposure, finance, and access to media has a lower likelihood of having SW. To reduce the disparity among poor and non-poor households in access to quality information and health education, it may be necessary to widen child nutrition programme, by engaging healthcare workers to facilitate education on the importance of good nutrition as well as consequences of poor nutrition. Such education intervention might be in the form of door to door activities and peer and social network mobilization.

The importance of maternal education in reducing the inequalities in SW should also be given prominence. Improving women education has been advocated both locally and globally as a channel of enhancing child health outcomes, especially in LMICs [8,16,18]. We found maternal age as an important contributor to poverty inequality in SW distribution with higher risk among children with poor mothers than those of non-poor mothers. This might have been affected by the societal values and disapprovals associated with childbearing outside marriage [40]. Such may negatively affect the type of support and help offered to mothers and their children. A special intervention focussing on mothers with no education should be put in place so that the poverty-related inequalities in the distribution of SW can be eliminated.

## Study limitations and strengths

The variations in the compositional and structural effects of the factors associated with poverty inequality in SW across the countries showed that different factors are specific to each country. Some of these factors, such as economic and political instability, war, famine, conflict and climate change, are outside the scope of the current study. This is one of our study limitations. Also, Blinder-Oaxaca decomposition does not address causality but rather quantifies contributions of associated factors to inequalities. Nonetheless, our study has strengths. We have used nationally representative data involving over half of a million in 51 countries. Our findings are generalizable in all the countries involved in this study. LMICs should put in place multi-sectoral country-specific intervention to ease the burden of SW. This intervention is very important as the cultural and social barriers faced by different population sub-groups can adversely affect health outcomes with dire consequences for their health, which may further perpetuate their disproportionate levels of poverty and lead to cycles of poverty [2].

## Conclusion

This study identified a wide gap between the propensity of children from poor and non-poor households to develop severe wasting. We decomposed the determinants of this crucial health outcome into two groups based on the wealth quintiles of the households from which the children come from. While different determinants are specific to different countries both in the compositional and structural components, some determinants are specific to certain neighbourhoods. Neighbourhood socioeconomic disadvantage, media access, as well as maternal age and maternal educational attainment created widest gaps in the inequalities between the children from poor and non-poor households in developing SW.

## Policy and program implications

Poverty, the principal cause of malnutrition must be tackled headlong, especially in the pro-poor countries. There is a need for a policy on education for the populace, especially for the women, as well as on the reduction of unemployment and enhancement of means of livelihoods. Combating poverty inequality in the development of severe wasting is a war that could only be won if confronted with multi-sectoral and country-specific interventions in low- and middle-income countries with considerations for the factors identified in this study. An efficient and effective severe wasting prevention strategies will aid healthy living, lower opportunity infections and reduce childhood mortalities and thereby contribute to the attainment of the SDG 3.

There are needs for the stakeholders and government of the countries with high pro-poor inequality and high prevalence of severe wasting to design policies and programs aimed at simultaneously lowering the occurrence of severe wasting and reducing socioeconomic inequalities among children from poor and non-poor households. These countries may need

to understudy what has been done in countries with lower prevalence and low inequalities. Whereas the countries with high rates of severe wasting and high pro-non-poor inequalities should formulate and implement policies aimed at lowering the prevalence while necessary education on children diets should be in place. Also, there are needs for policies and programs targeted at reducing pro-poor inequalities in the countries with high pro-poor inequality but low prevalence of severe wasting. There is a need too for countries with low severe wasting and high pro-non-poor inequality to develop policies targeted at the households in the richer wealth quintiles to embrace better feeding habits for under-five children.

## Implications for future research

While this study is a good start in identifying factors that contribute to socioeconomic inequalities in severe wasting, there are needs for further dialogue and research about social and cultural issues that may be associated with severe wasting. A qualitative study may help elucidate these. Besides, it may be necessary to study what is been done right in those countries with a low prevalence of severe wasting and low-risk differences and the lessons learnt can be adopted in countries with a high prevalence of severe wasting and high-risk differences. Also, there is a need to research into the factors that contributed to pro-non-poor inequalities in severe wasting in Chad, Egypt and Tajikistan.

## Supporting information

**S1 Table. Detailed decomposition analysis.**
(DOCX)

## Acknowledgments

The authors are grateful to ICF Macro, USA, for granting the authors the request to use the DHS data.

## Author Contributions

**Conceptualization:** Adeniyi Francis Fagbamigbe, Olalekan A. Uthman.

**Data curation:** Adeniyi Francis Fagbamigbe.

**Formal analysis:** Adeniyi Francis Fagbamigbe.

**Investigation:** Adeniyi Francis Fagbamigbe, Ngianga-Bakwin Kandala, Olalekan A. Uthman.

**Methodology:** Adeniyi Francis Fagbamigbe.

**Resources:** Adeniyi Francis Fagbamigbe.

**Supervision:** Ngianga-Bakwin Kandala, Olalekan A. Uthman.

**Writing – original draft:** Adeniyi Francis Fagbamigbe.

**Writing – review & editing:** Adeniyi Francis Fagbamigbe, Ngianga-Bakwin Kandala, Olalekan A. Uthman.

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
