## [Decision Letter · Decision Letter 0]

22 Jul 2020

PONE-D-20-04072

Mind the Gap: What explains the poor-non-poor inequalities in severe acute malnutrition among under-five children in Low- and Middle-Income countries? Compositional and structural characteristics

PLOS ONE

Dear Dr. Fagbamigbe,

Thank you for submitting your manuscript to PLOS ONE. Please accept my apologies for the delay in getting a decision to you. After careful consideration of the reports of two external reviewers, we feel that your article has merit but does not fully meet PLOS ONE’s publication criteria as it currently stands. Therefore, we invite you to submit a revised version of the manuscript that addresses the points raised during the review process.

Specifically, the reviewers raised concerns about some aspects of the methodology used in your study and requested that you include further discussion of some key points.

We look forward to receiving your revised manuscript.

Kind regards,

Dr Joseph Donlan

Staff Editor

PLOS ONE

Journal Requirements:

Additional Editor Comments (if provided):

Reviewers' comments:

Reviewer's Responses to Questions

**Comments to the Author**

1. Is the manuscript technically sound, and do the data support the conclusions?

Reviewer #1: Partly

Reviewer #2: Yes

2. Has the statistical analysis been performed appropriately and rigorously? 

Reviewer #1: Yes

Reviewer #2: Yes

3. Have the authors made all data underlying the findings in their manuscript fully available?

Reviewer #1: Yes

Reviewer #2: Yes

4. Is the manuscript presented in an intelligible fashion and written in standard English?

Reviewer #1: No

Reviewer #2: Yes

5. Review Comments to the Author

Reviewer #1: I am very glad to review this manuscript. It is a very important subject of global health. Above my suggestions:

I understand the WHO definition of SAM used in this paper. However, more recent references define malnutrition as related to both deficiencies and excesses in nutrition, and then, therefore, it includes wasting, stunting, underweight, micronutrient deficiencies or excesses, overweight, and obesity (see WHO Fact Sheet: https://www.who.int/news-room/fact-sheets/detail/malnutrition#:~:text=Malnutrition%20refers%20to%20deficiencies%2C%20excesses,low%20weight%2Dfor%2Dage)%3B). I think the title and the text will benefit if referring the outcome as severe wasting.

Introduction:

Line 53: The entire paragraph needs revision. It starts with “Irrespective of household wealth status, malnutrition is one of the inequalities in health outcomes among millions of children globally”. I guess it is trying to stablish that malnutrition is one of the health outcomes with higher level of inequality, is it correct? In addition, authors end the paragraph listing many risk factors of malnutrition among children, all of them associated with lower levels of wealth, despite what they established in line 53.

Line 80: Although the initial SAM was defined in the Abstract, the text will benefit of an additional definition here (as made for LMICs).

Line 91: Authors declared that “Findings from our study are potential ingredients for global, national and subnational policy and intervention in child nutrition.” It’s an ecological design, with data on a large number of surveys. I think it provides information to national policies, however, several within-country inequalities may exist and it may hinder sub-national validation.

Methodology

1. What criteria authors used to define countries region? Myanmar and Timor Leste are East Asia and Pacific according to UNICEF and South-East Asia according to WHO, not Caribbean.

2. Subsection of BODA explanation: since it is not a methological paper and being the BODA methodology available elsewhere, I think authors could rewrite this subsection focusing on their model instead of an extensive general approach. It will turn the reading and understanding of the article much easier and more adequate to PlosONE audience.

3. I do not think the inclusion criteria is clear enough. Why are only three countries in Latin America? I realize most data in the region is from MICS or RHS, however, there are DHS carried out since 2010 with data on anthropometry (for example Colombia 2010).

4. Independent variables were only cited. Please add the methods used to select this variables and introduce the importance of each variable to SAM and wealth-based inequality in SAM.

Results

Line 303: “Across the countries, there were variations in the effect of the factors associated with wealth inequalities. Hence, the decomposition analysis involved only 20 countries”. How these differences were identified? More information should be available in the supplementary material.

Line 314: Is “educational inequality” correct in this sentence? Instead of “socioeconomic inequality”, measured through DHS wealth index?

Discussion/Conclusion

Discutir resultado da decomposição para Timor Leste

This section needs revision. It is more establishing the results than discussing the more impressive results found.

I would like to see specially a couple of things more discussed:

1. An explanation or authors hypothesis regarding countries with pro-non-poor inequalities in SAM. It is a very surprising result, considering the high cutoff (-3SD);

2. Since Timor Leste is an outlier at both prevalence of SAM and according to decomposition analysis, text will benefit of a major focus on specific discussion about the country.

3. Thinking about policies and programs, I suggest a paragraph recommending policies to each group of countries according with definition in lines 294-27. For example, I understand that countries from group 3 (high pro-poor inequality with low prevalence) are in a better situation than countries from group 1 (high SAM and high pro-poor inequality), but since a smaller group still with SAM, it probably refers to a harder-to-reach subgroup which requires a more specific approach, different than group 1.

Important references to be cited:

https://healtheconomicsreview.biomedcentral.com/track/pdf/10.1186/s13561-016-0097-3 - Blinder-Oaxaca decomposition of child malnutrition in Egypt, Jordan and Yemen.

Reviewer #2: Many thanks to the authors, it is quite amazing to get a paper using such rarely used econometric

methods. The paper is interesting, but a few things need to be checked. My evaluation is as below.

• The introduction needs to be linked to SDGs on health. Otherwise, it is unclear which development issues they are addressing; they have written the introduction in a policy vacuum

• I am also concerned with the intuition behind mixing data from different regions say, Asia, Africa, Latin America and do one decomposition. Rather characteristics in these places are different and they ought to be done differently, for each region-my thought.

• On independent variables, beginning line 127, that whole thing is just one sentence. Places cut it properly and define those variables rather than just listing. Are these variables based on theory or empirical evidence to suggest that they may have an impact? Please cite.

• One limitation probably is the fact that the OB decomposition does not address causality; hence the results should be interpreted with caution. Please highlight this limitation

• There is a need to put implications for future research- this is missing in the paper

• Also, the study doesn’t provide any policy implications. They indicate that poverty should be tackled but does not say how it should be done. A sentence or two will be helpful.

• There are several typos, and the authors should read again to address these.

6. PLOS authors have the option to publish the peer review history of their article (what does this mean?). If published, this will include your full peer review and any attached files.

Reviewer #1: No

Reviewer #2: No

---

## [Author Response · Author response to Decision Letter 0]

6 Aug 2020

July 24th 2020

The Editor,

Plos One

PONE-D-20-04072

Mind the Gap: What explains the poor-non-poor inequalities in severe wasting among under-five children in low- and middle-income countries? compositional and structural characteristics

Dear editor, the authors appreciate your efforts and that of the reviewer of this manuscript for the interest to get the best out of the paper. We found all the comments very useful and insightful. We have responded to all the comments. A point-by-point response have been made to all the issues raised as stated below and all necessary changes have been made in the revised version.

Reviewer #1: I am very glad to review this manuscript. It is a very important subject of global health. 

THANK YOU

Above my suggestions:

I understand the WHO definition of SAM used in this paper. However, more recent references define malnutrition as related to both deficiencies and excesses in nutrition, and then, therefore, it includes wasting, stunting, underweight, micronutrient deficiencies or excesses, overweight, and obesity (see WHO Fact Sheet: https://www.who.int/news-room/fact-sheets/detail/malnutrition#:~:text=Malnutrition%20refers%20to%20deficiencies%2C%20excesses,low%20weight%2Dfor%2Dage)%3B). I think the title and the text will benefit if referring the outcome as severe wasting.

THANK YOU FOR THE SUGGESTION. WE HAVE ADOPTED THE DEFINITION AND CHANGED THE TITLE. SEE THE CHANGES ACROSS THE PAPER

Introduction:

Line 53: The entire paragraph needs revision. It starts with “Irrespective of household wealth status, malnutrition is one of the inequalities in health outcomes among millions of children globally”. I guess it is trying to stablish that malnutrition is one of the health outcomes with higher level of inequality, is it correct? In addition, authors end the paragraph listing many risk factors of malnutrition among children, all of them associated with lower levels of wealth, despite what they established in line 53.

THANK YOU. WE HAVE REVISED THE PARAGRAPH ACCORDINGLY. SEE LINES 65-72 in the tracked file and Lines 73-81 in the clean manuscript file.

Line 80: Although the initial SAM was defined in the Abstract, the text will benefit of an additional definition here (as made for LMICs).

THANK YOU. WE HAVE PROVIDED THE DEFINITIONS. SEE LINES 50,53 and 95 in the tracked file and Lines 61, 64 in the clean manuscript file.

Line 91: Authors declared that “Findings from our study are potential ingredients for global, national and subnational policy and intervention in child nutrition.” It’s an ecological design, with data on a large number of surveys. I think it provides information to national policies, however, several within-country inequalities may exist and it may hinder sub-national validation.

THANK YOU, WE HAVE REVERESED THE SENTENCE SEE LINES 108 in the tracked file and Lines 118 in the clean manuscript file.

Methodology

1. What criteria authors used to define countries region? Myanmar and Timor Leste are East Asia and Pacific according to UNICEF and South-East Asia according to WHO, not Caribbean.

THANK YOU, WE AGREE WITH THIS OMISSION, WE USED THE WHO CLASSIFICATIONS AND REANALYSED THE DATA ACCORDINGLY SEE TABLE 1 AND FIGURE 1

2. Subsection of BODA explanation: since it is not a methological paper and being the BODA methodology available elsewhere, I think authors could rewrite this subsection focusing on their model instead of an extensive general approach. It will turn the reading and understanding of the article much easier and more adequate to PlosONE audience.

THANK YOU, WE WERE REQUESTED EARLIER TO PROVIDE A STATISTICAL DETAIL OF THE METHODOLOGY. PLOS ONE REQUESTS FOR STATISTICAL DETAILS OF METHODS USED. WE ARE COMPELLED TO RETAIN THE DETAILS BECAUSE THE SECOND EXTERNAL REVIEWERS DID NOT RECOMMEND A DELETION OF THE SECTION

3. I do not think the inclusion criteria is clear enough. Why are only three countries in Latin America? I realize most data in the region is from MICS or RHS, however, there are DHS carried out since 2010 with data on anthropometry (for example Colombia 2010).

WE WORKED WITH OUR STATED INCLUSION CRITERIA. ”THE MOST RECENT DHS DATA BETWEEN 2010 AND 2018 DATA ON ANTHROPOMETRY”. THE COLOMBIA 2015 WAS THE MOST RECENT DATA IN COLOMBIA. WE EXPLORED THE 2015 DATA BUT IT CONTAINED NO DATA ON ANTHROPOMETRY. SEE LINES 113-116 in the tracked file and Lines 124-128 in the clean manuscript file.

4. Independent variables were only cited. Please add the methods used to select this variables and introduce the importance of each variable to SAM and wealth-based inequality in SAM.

WE HAVE PROVIDE CITATION AND HOW THE INDEPENDENT VARIABLES WERE SEARCHED AND IDENTIFIED. SEE LINES 149-180 in the tracked file and Lines 161-183 in the clean manuscript file.

Results

Line 303: “Across the countries, there were variations in the effect of the factors associated with wealth inequalities. Hence, the decomposition analysis involved only 20 countries”. How these differences were identified? More information should be available in the supplementary material.

THANK YOU. WE HAVE NOW PROVIDED A SUPPLEMENTARY ANALYSIS SHOWING THE DETAILED DECOMPOSITION ANALYSIS. SEE LINE 347 in the tracked file and Lines 376 in the clean manuscript file.AND THE SUPPLEMENTARY MATERIAL

Line 314: Is “educational inequality” correct in this sentence? Instead of “socioeconomic inequality”, measured through DHS wealth index?

THANK YOU. IT HAS BEEN CORRECTED SEE LINES 342 in the tracked file and Lines 369 in the clean manuscript file.

Discussion/Conclusion

Discutir resultado da decomposição para Timor Leste

YES, WE HAVE DISCUSSED THE RESULTS OF THE TIMOR-LETSE DECOMPOSITION ANALYSIS. SEE LINES 469-478 in the tracked file and Lines 516-525 in the clean manuscript file.

This section needs revision. It is more establishing the results than discussing the more impressive results found.

WE HAVE REVISED THE DISCUSSION AND PROVIDED MORE CITATIONS SEE LINES 383 – 514 in the tracked file and Lines 414-552 in the clean manuscript file.

I would like to see specially a couple of things more discussed:

1. An explanation or authors hypothesis regarding countries with pro-non-poor inequalities in SAM. It is a very surprising result, considering the high cutoff (-3SD);

WE AGREE WITH THIS AND HAVE PROVIDED OUR HYPOTHESIS IN THE INTRODUCTION AND PROVIDED PLAUSIBLE REASONS FOR OUR RESULTS. SEE LINES 419-440 in the tracked file and Lines 461 – 483 in the clean manuscript file.

2. Since Timor Leste is an outlier at both prevalence of SAM and according to decomposition analysis, text will benefit of a major focus on specific discussion about the country.

WE AGREE WITH THIS COMMENT. WE HAVE DISCUSSED THE RESULTS OF THE TIMOR-LETSE DECOMPOSITION ANALYSIS. SEE LINES 469-478 in the tracked file and Lines 516-525 in the clean manuscript file.

3. Thinking about policies and programs, I suggest a paragraph recommending policies to each group of countries according with definition in lines 294-27. For example, I understand that countries from group 3 (high pro-poor inequality with low prevalence) are in a better situation than countries from group 1 (high SAM and high pro-poor inequality), but since a smaller group still with SAM, it probably refers to a harder-to-reach subgroup which requires a more specific approach, different than group 1.

WE HAVE DISCUSSED THE POLICY IMPLICATIONS FOR THE FOUR SCENARIOS. SEE LINES 524-544 in the tracked file and Lines 577-598 in the clean manuscript file.

Important references to be cited:

https://healtheconomicsreview.biomedcentral.com/track/pdf/10.1186/s13561-016-0097-3 - Blinder-Oaxaca decomposition of child malnutrition in Egypt, Jordan and Yemen.

THANK YOU FOR PROVIDING THIS IMPORTANT REFERENCE. IT IS NOW OUR REFERENCE 41.

Reviewer #2: Many thanks to the authors, it is quite amazing to get a paper using such rarely used econometric

methods. The paper is interesting, but a few things need to be checked. My evaluation is as below.

THANK YOU

• The introduction needs to be linked to SDGs on health. Otherwise, it is unclear which development issues they are addressing; they have written the introduction in a policy vacuum

WE AGREE WITH THE COMMENT. WE HAVE PROVIDED THE RELEVANT SDG. SEE LINES 50-53 in the tracked file and Lines 58-61 in the clean manuscript file.

• I am also concerned with the intuition behind mixing data from different regions say, Asia, Africa, Latin America and do one decomposition. Rather characteristics in these places are different and they ought to be done differently, for each region-my thought.

THANK YOU, THE DATA WERE NOT MIXED. ALSO, WE DID NOT DO ONE DECOMPOSITION. RATHER, THE DECOMPOSITION ANALYSIS WERE COUNTRY-SPECIFIC. ONLY THE 20 COUNTRIES THAT HGAD PRO-POOR INEQUALITIES IN SEVERE WASTING WERE CANDIDATES FOR THE DECOMPOSITION ANALYSIS. THE ANALYSIS WERE THEN CARRIED OUT WITHIN EACH OF THE 20 COUNTRIES ONE AFTER THE OTHER AS SHOWN IN THE SUPPLEMENTARY FILE S1 Tables.

• On independent variables, beginning line 127, that whole thing is just one sentence. Places cut it properly and define those variables rather than just listing. Are these variables based on theory or empirical evidence to suggest that they may have an impact? Please cite.

THE VARIABLES WERE IDENTIFIED EMPIRICALLY AS THEY WERE REPORTED TO HAVE IMPACT ON THE STUDY OUTCOME. WE HAVE PROVIDED ADDITIONAL INFORMATION AND RE-FORMATTED THE PARAGRAPH. SEE LINES 150-180 in the tracked file and Lines 161-183 in the clean manuscript file.

• One limitation probably is the fact that the OB decomposition does not address causality; hence the results should be interpreted with caution. Please highlight this limitation

THE AUTHORS AGREE WITH OUR REVIEWER, WE HAVE ADDED THIS LIMITATION SEE LINES 502-514 in the tracked file and Lines 558 -562 in the clean manuscript file.

• There is a need to put implications for future research- this is missing in the paper

 THANK YOU. WE HAVE IDENTIFIED AND SUGGESTED AREAS OF FUTURE RESEARCH. SEE LINES 545-553 in the tracked file and Lines 599-607 in the clean manuscript file.

• Also, the study doesn’t provide any policy implications. They indicate that poverty should be tackled but does not say how it should be done. A sentence or two will be helpful.

THE AUTHORS HAVE ADDED POLICY IMPLICATIONS. SEE LINES 524-544 in the tracked file and Lines 577-598 in the clean manuscript file.

• There are several typos, and the authors should read again to address these. 

THANK YOU. WE HAVE COPY-EDITED THE MANUSCRIPT AS DIRECTED AND CORRECTED TYPOS

Signed

Adeniyi Fagbamigbe

---

## [Decision Letter · Decision Letter 1]

15 Oct 2020

Mind the Gap: What explains the poor-non-poor inequalities in severe wasting among under-five children in Low- and Middle-Income countries ? Compositional and structural characteristics

PONE-D-2 0-04072R1

Dear Dr. Fagbamigbe,

We’re pleased to inform you that your manuscript has been judged scientifically suitable for publication and will be formally accepted for publication once it meets all outstanding technical requirements.

Kind regards,

Akihiro Nishi, M.D., Dr.P.H.

Academic Editor

PLOS ONE

Additional Editor Comments (optional):

I am pleased to accept the paper! Please fix some typos upon the proof-reading.

Reviewers' comments:

Reviewer's Responses to Questions

**Comments to the Author**

1. If the authors have adequately addressed your comments raised in a previous round of review and you feel that this manuscript is now acceptable for publication, you may indicate that here to bypass the “Comments to the Author” section, enter your conflict of interest statement in the “Confidential to Editor” section, and submit your "Accept" recommendation.

Reviewer #1: All comments have been addressed

Reviewer #2: All comments have been addressed

2. Is the manuscript technically sound, and do the data support the conclusions?

Reviewer #1: Yes

Reviewer #2: Yes

3. Has the statistical analysis been performed appropriately and rigorously? 

Reviewer #1: Yes

Reviewer #2: Yes

4. Have the authors made all data underlying the findings in their manuscript fully available?

Reviewer #1: Yes

Reviewer #2: Yes

5. Is the manuscript presented in an intelligible fashion and written in standard English?

Reviewer #1: Yes

Reviewer #2: Yes

6. Review Comments to the Author

Reviewer #1: (No Response)

Reviewer #2: I have gone point by point regarding the things I asked them to do. I am glad to say that they have addressed all the things. A job well done. However, they should just go through the paper again, to remove small typos

7. PLOS authors have the option to publish the peer review history of their article (what does this mean?). If published, this will include your full peer review and any attached files.

Reviewer #1: No

Reviewer #2: No

---

## [Editor Report · Acceptance letter]

22 Oct 2020

PONE-D-20-04072R1 

Mind the gap: what explains the poor-non-poor inequalities in severe wasting among under-five children in low- and middle-income countries? compositional and structural characteristics 

Dear Dr. Fagbamigbe:

I'm pleased to inform you that your manuscript has been deemed suitable for publication in PLOS ONE. Congratulations! Your manuscript is now with our production department. 

Kind regards, 

on behalf of

Dr. Akihiro Nishi 

Academic Editor

PLOS ONE